# IoT in Water Quality Monitoring—Are We Really Here?

**DOI:** 10.3390/s23020960

**Published:** 2023-01-14

**Authors:** Małgorzata Miller, Anna Kisiel, Danuta Cembrowska-Lech, Irmina Durlik, Tymoteusz Miller

**Affiliations:** 1Polish Society of Bioinformatics and Data Science BIODATA, 71-214 Szczecin, Poland; 2Institute of Marine and Environmental Science, University of Szczecin, 71-415 Szczecin, Poland; 3Institute of Biology, University of Szczecin, 71-412 Szczecin, Poland; 4Faculty of Navigation, Maritime University of Szczecin, 70-500 Szczecin, Poland

**Keywords:** IoT, water quality, efficiency

## Abstract

The Internet of Things (IoT) has become widespread. Mainly used in industry, it already penetrates into every sphere of private life. It is often associated with complex sensors and very complicated technology. IoT in life sciences has gained a lot of importance because it allows one to minimize the costs associated with field research, expeditions, and the transport of the many sensors necessary for physical and chemical measurements. In the literature, we can find many sensational ideas regarding the use of remote collection of environmental research. However, can we fully say that IoT is well established in the natural sciences?

## 1. Introduction

The Internet of things (IoT) brings an exciting new dimension to the world of field research, where researchers are able to access their data and insights in real time, wherever they are. This means that physical and chemical measurements can be made on-site or in the lab with a fraction of the effort that it would take if one were to rely on manual data collection [1,2,3].

It also means that one can deploy a network of sensors across a region of interest and have them all communicate with each other to provide a comprehensive picture of what is happening in that area. This is particularly valuable for water quality monitoring, where changes in conditions can have serious consequences downstream [3,4,5].

In recent years, the Internet of Things (IoT) has become increasingly popular for a wide range of applications, including monitoring the quality of water. By using IoT devices such as Raspberry Pi and sensors that measure various parameters such as temperature, oxygen, and pH, it is possible to continuously monitor the quality of water in real-time. This information can be collected and analyzed using programming languages such as Python and Julia to gain insights and make informed decisions about water management [1,3,4,5].

For example, a system that includes temperature sensors and oxygen sensors could be used to monitor the health of a lake or river. By detecting changes in temperature and oxygen levels, it is possible to identify potential issues such as algal blooms or changes in the ecosystem. Similarly, pH sensors and BOD sensors (biochemical oxygen demand) can be used to measure the acidity and pollution levels of water bodies [6,7,8,9].

The benefits of IoT in water quality monitoring are clear, but there are also some challenges that need to be considered. One of the biggest challenges is data management, as the volume of data generated by a large network of sensors can be enormous. It is also important to ensure that the data is of high quality, as this is essential for making accurate predictions about future conditions [10,11,12,13].

Another challenge is security, as it is vital to ensure that the data collected by the sensors is not accessed by unauthorized individuals. This is particularly important given the sensitive nature of water quality data. The Internet of things (IoT) brings an exciting new dimension to the world of field research, where researchers are able to access their data and insights in real time, wherever they are. This means that physical and chemical measurements can be made on-site or in the lab with a fraction of the effort that it would take if you were to rely on manual data collection [5,9,14].

It also means that you can deploy a network of sensors across a region of interest and have them all communicate with each other to provide a comprehensive picture of what is happening in that area. This is particularly valuable for water quality monitoring, where changes in conditions can have serious consequences downstream [5,9,14].

Despite these challenges, IoT in water quality monitoring is already starting to make a difference. In many cases, it is providing insights that would not have been possible without it. As the technology continues to develop, it is likely that its impact will only grow.

The purpose of this work is to introduce the concepts of the Internet of Things (IoT) in determining the quality of water and what factors it can be associated with.

## 2. Internet of Things Applications

The Internet of Things has been a buzzword in the tech industry for years now, but it seems like its applications are only just beginning to be fully realized. One area where the IoT is starting to have a major impact is in water quality monitoring [15,16,17,18,19,20,21].

There are many potential benefits to using the IoT in water quality monitoring. For instance, it can help to provide real-time data that can be used to make decisions about water usage and treatment. This is especially important in areas where water shortages are a concern. Additionally, the IoT can help to identify and track sources of pollution, which is essential for protecting public health and the environment.

Water quality monitoring is an important application of the IoT, but it is only one of many. The IoT has the potential to revolutionize many industries and change the way we live our lives. One potential application of IoT in water quality assessment is the use of sensors to monitor the levels of various contaminants and pollutants in water sources. These sensors can be placed in rivers, lakes, and other bodies of water, and they can collect data on a range of parameters, including pH levels, temperature, dissolved oxygen, and the presence of chemicals and microorganisms. This data can be transmitted in real-time to a central monitoring system, where it can be analyzed and used to identify potential issues and take corrective action if necessary [15,16,17,18,19,20,21].

Another potential use of IoT in water quality assessment is the deployment of smart water meters and other devices in homes and businesses. These devices can monitor water usage, detect leaks, and provide early warning of potential issues, such as the presence of lead or other contaminants in the water supply. By providing real-time data on water usage and quality, these devices can help water utilities to better manage their systems and respond quickly to potential problems.

The usage of IoT in water quality assessment may help to improve the safety and reliability of drinking water supplies by providing real-time monitoring and data collection capabilities. By enabling water utilities and other organizations to detect and respond to potential issues quickly and efficiently, IoT can help to ensure that communities have access to clean, safe, and sustainable water sources [15,16,17,18,19,20,21]. 

The most common water quality indicators that can be monitored and managed remotely using IoT technology are:pH: a measure of its acidity or basicity. It is important to monitor pH levels because they can affect the solubility and toxicity of certain substances, as well as the overall health of aquatic ecosystems.Temperature: water temperature can affect the solubility and toxicity of certain substances, as well as the metabolism and behavior of aquatic organisms.Dissolved oxygen: an important indicator of the health of an aquatic ecosystem. It is essential for the survival of most aquatic organisms, and low levels of dissolved oxygen can lead to fish dying and other problems.Total dissolved solids (TDS): a measure of the total amount of dissolved minerals, salts, and other substances in water. High levels of TDS can affect the taste and appearance of water, as well as the health of aquatic ecosystems.Turbidity: a measure of the clarity of water. High levels of turbidity can indicate the presence of sediment, algae, or other substances in the water, which can affect its quality.Conductivity: conductivity as a measure of the ability of water to conduct electricity. It can be used to infer the presence of certain ions in the water, which can affect its quality.Eh: also known as redox potential, is a measure of the oxidation-reduction potential of a solution. It is typically measured in millivolts (mV) and can be used to infer the presence of certain ions in the water, which can affect its quality.Chlorophyll a: used as an indicator of the presence of algae in water, which can affect its quality.COD: chemical oxygen demand, a measure of the amount of oxygen required to oxidize the organic matter in water. High levels of COD can indicate the presence of a large amount of organic matter in the water, which can lead to problems such as reduced water quality and the production of harmful by-products.Ammonia nitrogen: may be toxic to aquatic life at high levels, so it is important to monitor its concentration to ensure the health of aquatic ecosystems.

As an example of high-level flow chart of how an IoT system for water monitoring might work, the sequence of information flow may look like the following (Figure 1):Sensors: Water quality sensors are installed in the water sources being monitored (e.g., lakes, rivers, etc.). These sensors collect data on various parameters such as pH, temperature, and dissolved oxygen levels.Data transmission: The sensor data is transmitted to a central server or cloud service using a wireless communication protocol such as WiFi, Bluetooth, or cellular.Data storage: The sensor data is stored on the central server or cloud service.Data processing: The sensor data is processed and analyzed by algorithms or software to identify trends and patterns.Alerts: If the sensor data indicates that there is a problem with the water quality (e.g., pH levels are too high), an alert is sent to the appropriate authorities (e.g., the local water treatment plant).Response: The authorities can then take action to address the issue (e.g., adjust the pH levels at the water treatment plant).Data visualization: The sensor data can also be visualized using dashboards or other tools, allowing stakeholders to see the current water quality and trends over time.Data sharing: data collected and processed from IoT sensors may be transferred to the public, relevant authorities, non-governmental organizations or responsible persons.

## 3. Who Needs This and How It Benefits Them

There are many benefits to using IoT in water quality monitoring. For instance, it can help save money on water treatment and pumping costs. Additionally, it can improve the efficiency of water distribution, leading to less water waste and lower energy bills. In addition, IoT can provide real-time data on water quality, allowing for more timely interventions when problems are detected. This can lead to improved public health and safety, as well as reduced environmental impacts from water contamination. Finally, deploying IoT in water quality monitoring can help build public trust in the government and utilities that manage our water resources. Some potential issues that may arise when integrating IoT technology into the water management process include:Public water governance companies may not have the necessary skills or infrastructure in place to effectively manage an IoT system.Public water governance companies may be concerned about the cost of implementing and maintaining an IoT system.Privacy and ownership of data: There may be concerns regarding who owns and protects the data collected by IoT sensorsThere may be challenges in integrating the IoT system with the water governance company’s existing systems and processes.Water management may be subject to regulatory challenges or concerns related to IoT technology.

To address these challenges and ensure a smooth integration of the technology, both the public water governance company and the provider of IoT technology need to work together. For all concerns to be addressed, stakeholders such as local communities and regulatory bodies may also need to be included in the process.

## 4. Common Solutions

In general, there are several common IoT technologies that are used for water quality estimation. The summarized list includes [22,23,24,25,26,27]: (a)Sensors: IoT sensors are used to measure various parameters of water quality, such as pH, temperature, dissolved oxygen, and the presence of chemicals and microorganisms. These sensors can be placed in rivers, lakes, and other bodies of water, and they can transmit data in real-time to a central monitoring system.(b)Smart meters: IoT smart meters are used to monitor water usage in homes and businesses, and they can detect leaks and other potential issues. By providing real-time data on water usage, these devices can help water utilities to better manage their systems and respond to potential problems.(c)Actuators: IoT actuators are devices that can control or manipulate the environment based on data from sensors. In the context of water quality, actuators can be used to automatically adjust the flow of water, or to dispense chemicals or other substances to improve the quality of the water.(d)Gateways: IoT gateways are devices that can connect multiple sensors and other devices to the internet, and they can act as intermediaries between the devices and a central monitoring system. In the context of water quality, gateways can be used to collect data from multiple sensors and transmit it to a central server for analysis and processing.

A table with examples of the shown list is presented in Appendix A.

The Raspberry Pi is a popular choice for the use in IoT projects, including those involving water monitoring. The Raspberry Pi is a small, inexpensive computer that is well-suited for use in IoT applications because of its low power consumption, small size, and ability to run a variety of software.

There are many other servers and devices that can also be used for analyzing and processing data in an IoT system for water monitoring. Some other options might include:Industrial PCs are ruggedized computers that are designed for use in industrial environments. They are often more expensive than the Raspberry Pi but offer more processing power and other features that may be useful in certain applications. They are often ruggedized to withstand harsh conditions, such as extreme temperatures, vibration, dust and humidity. Industrial PCs are used in a variety of applications, including manufacturing, transportation, and energy production.Cloud services, such as Microsoft Azure or Amazon Web Services, can be used to store and process data in an IoT system. This can be a convenient option because it eliminates the need to set up and maintain physical servers.Microcontrollers, such as the Arduino, are small, low-power computers that can be used to collect data from sensors and transmit it to a central server or cloud service. They are often less expensive than other options and are well-suited for use in small, simple IoT projects.Custom hardware can be developed specifically for use in an IoT system for water quality monitoring. This can be a more expensive option but may be necessary in certain cases where off-the-shelf hardware does not meet the necessary requirements. Custom hardware refers to hardware that is specifically designed and built for a particular application or use case. In the context of an IoT system for water quality monitoring, custom hardware could refer to sensors, microcontrollers, or other devices that are designed and built specifically for use in the system.

Custom hardware is often developed when off-the-shelf hardware is not available or does not meet the necessary requirements for the application. For example, a company might develop custom hardware if it needs sensors with specific performance characteristics or if it needs to integrate the hardware into a larger system with unique requirements. Developing custom hardware can be a more expensive option than using off-the-shelf hardware, but it may be necessary in certain cases where the necessary features or capabilities are not available in commercial products.

Because of low cost and relatively easy usage the common practice is to create one’s own solutions using Raspberry Pi. The Raspberry Pi (RP) is a small, low-cost computer that is well-suited for use in Internet of Things (IoT) projects, including projects related to water quality estimation. To use a Raspberry Pi for water quality estimation with IoT, some steps can be made: (A) Installation of the necessary software on the Raspberry Pi, including an operating system such as Raspbian, and any libraries or tools that may be needed for project, such as Python or Julia for programming, and libraries for interfacing with sensors or other hardware. (B) Connection of the Raspberry Pi to the internet, either via a wired or wireless connection, so that it can communicate with other devices and services as part of an IoT network. (C) Attachment of the appropriate sensors to the Raspberry Pi, such as pH, Eh, and oxygen electrodes, to measure the quality of the water. (D) Usage of the Raspberry Pi’s connectivity and computing capabilities to collect and transmit data from the sensors in real-time. Obtained data can be sent to a central server or cloud service for analysis and processing, or it can be used directly by the Raspberry Pi for local decision-making and control. (E) Usage of the data collected by the Raspberry Pi and its sensors to monitor the quality of the water and identify any potential issues or concerns. This can include alerting operators or other stakeholders if certain thresholds are exceeded, or automatically taking corrective action if necessary [28,29,30,31,32].

There are a few potential disadvantages to using a Raspberry Pi for IoT and water quality estimation. These include: (1) Limited computing power: The Raspberry Pi is a relatively low-power device, and it may not be able to handle complex or resource-intensive tasks, such as advanced data analysis or machine learning algorithms. (2) Limited connectivity options: The Raspberry Pi may not support all forms of connectivity, such as cellular or satellite, which can limit its ability to communicate with other devices or services in an IoT network. (3) Limited sensor support: The Raspberry Pi may not have built-in support for certain sensors, such as pH, Eh, or oxygen electrodes, and usage of additional hardware or software to interface with these sensors may be needed. (4) Limited reliability: The Raspberry Pi is a single-board computer, and it is vulnerable to damage or failure if it is not properly handled or maintained. This can potentially compromise the integrity and reliability of the data collected by the system [30,31,32].

Example code of Python 3 and Julia to program the Raspberry Pi are presented in Appendix A. 

## 5. Challenges for IoT in Water Quality Monitoring

The biggest challenge for projecting an IoT system for water quality monitoring involves a number of steps. The first step in projecting an IoT system for water quality monitoring is to identify the specific parameters that need to be measured. This will depend on the specific needs of the application and the regulatory requirements. It is important to consider the possibility of emergency situations when designing an IoT system for water quality monitoring. In the case of a surface water system that is organized in a hierarchical manner, it may be appropriate to use fixed sensors at key locations to monitor the water quality. However, it is also important to have the ability to quickly deploy mobile sensors in the event of an emergency, such as a spill or other contamination event. The next step is to select the sensors and other hardware that will be used to collect and transmit data. This may include sensors to measure various water quality parameters, as well as microcontrollers or other devices to collect and transmit the data. 

Another is the communication and networking infrastructure that will be used to transmit data from the sensors to a central server or cloud service. This may include the use of wireless technologies such as WiFi, Bluetooth or cellular, as well as wired connections. The software and algorithms used to process and analyze the data collected by the sensors will be an important part of the IoT system. This may include the use of machine learning algorithms to identify trends and patterns in the data.

Once the IoT system has been designed and developed, it will need to be tested to ensure that it is functioning properly. After testing is complete, the system can be deployed in the field to begin collecting and transmitting data.

IoT technologies offer a number of benefits for water quality monitoring, including the ability to continuously collect data, the ability to remotely access and analyze data, and the ability to quickly respond to any issues that may arise. With the help of these technologies, it is possible to more effectively manage and protect the quality of our water resources [15,16,17,18,19,20,21].

Continuous monitoring: IoT devices can continuously collect data on various parameters such as temperature, pH, and oxygen levels, providing a constant stream of information about the quality of water. This allows for more frequent and comprehensive assessments than would be possible using traditional methods such as manual sampling and analysis.

Remote access and analysis: IoT devices can transmit data wirelessly, allowing for remote access and analysis of water quality information. This is particularly useful for monitoring water quality in remote or hard-to-reach locations.

Quick response to issues: By continuously monitoring water quality, it is possible to quickly identify and respond to any issues that may arise. This can help to prevent problems from becoming more serious and costly to address.

Increased efficiency: Using IoT technologies can help to streamline the process of water quality assessment, allowing for more efficient and cost-effective monitoring. One of the main challenges for IoT in water quality monitoring is the lack of standardization. There are many different types of sensors and devices on the market, and it can be difficult to find compatible products that work together. In addition, there is a lack of interoperability between different systems, which makes it difficult to exchange data between different devices. Another challenge is the cost of installing and maintaining an IoT system. Sensors and devices can be expensive, and often need to be replaced or upgraded regularly. Furthermore, IoT systems require ongoing maintenance and support, which can add to the overall cost. Finally, another challenge for IoT in water quality monitoring is ensuring the security and privacy of data. Because IoT systems collect and store large amounts of sensitive data, there is a risk that this data could be accessed by unauthorized individuals or organizations. As such, it is important to implement security measures to protect this data from unauthorized access [15,16,17,18,19,20,21].

Many potential disadvantages to using IoT in water quality assessment can be found. One concern is the cost of implementing and maintaining an IoT system, which can be significant. This may be particularly challenging for smaller water utilities or organizations with limited budgets [15,16,17,18,19,20,21].

Another potential disadvantage is the risk of cyberattacks or other security breaches. Because IoT systems rely on networks of connected devices and sensors, they can be vulnerable to hacking and other forms of cybercrime. This can potentially compromise the security and integrity of the data collected by the system, and it may also expose sensitive information about water sources and infrastructure to unauthorized parties. One example of a cyberattack on IoT is the Mirai botnet attack, which occurred in 2016. The Mirai botnet was a network of infected devices, such as security cameras and routers, which were used to launch distributed denial of service (DDoS) attacks against a number of high-profile websites and internet services. The attackers were able to take control of the infected devices by exploiting vulnerabilities in their software and using them to overwhelm the targeted websites with traffic, rendering them inaccessible to users. Another example is the Stuxnet worm, which was discovered in 2010. The Stuxnet worm was a piece of malware that was specifically designed to target industrial control systems, such as those used in power plants and other critical infrastructure. The worm was able to infect and take control of the systems, allowing the attackers to manipulate their operation and potentially cause damage or disruption. These are just a few generally known examples of the types of cyberattacks that can be carried out against IoT devices. While the potential for such attacks is a concern, there are steps that can be taken to mitigate the risks, such as implementing strong security measures and regularly updating software and firmware to address known vulnerabilities. Finally, there is the potential for human error or misuse of IoT data. The data collected by an IoT system is only as good as the people and processes that are responsible for interpreting and acting on it. If the data is not analyzed properly, or if it is used inappropriately, it can lead to incorrect decisions or actions that may negatively impact water quality [14,17,33,34,35,36,37].

While the use of IoT in water quality assessment can provide many benefits, it is important to carefully consider the potential disadvantages and take steps to mitigate any potential risks.

Ensuring the security of data in an IoT system is a critical concern, as it is important to protect the confidentiality, integrity, and availability of the data. There are a number of measures that can be taken to ensure the security of data in an IoT system:Data confidentiality: To protect the confidentiality of data, it is important to ensure that only authorized users have access to the data. This can be achieved through the use of encryption and secure authentication methods.Data integrity: To ensure the integrity of data, it is important to prevent unauthorized changes to the data and to detect if any changes have been made. This can be achieved through the use of cryptographic hashing and other techniques.Data availability: To ensure the availability of data, it is important to ensure that the data is accessible to authorized users when needed. This can be achieved through the use of redundant systems and other measures to prevent downtime.

There are a number of security functions available for IoT systems, including encryption, authentication, and access control. These functions can be implemented at various points in the system, such as at the sensor level, at the network level, or at the server level. It is important to consider the security needs of the specific application and to implement appropriate security measures throughout the system.

Using secure passwords:Keeping devices and software up to date: keeping all of devices and software in IoT systems up to date with the latest security patches. This can help prevent vulnerabilities from being exploited.Encryption usage to secure the communication between devices and any servers or cloud services that are connected to IoT systems.

There are many types of encryptions that can use to secure the communication between IoT devices and any servers. The most commonly used are:SSL (Secure Sockets Layer) and TLS (Transport Layer Security) are protocols that are commonly used to encrypt communication over the internet. They are typically used to secure websites and other internet-based services.IPSec (Internet Protocol Security) is a protocol that can be used to encrypt communication between devices over a network. It is often used to secure communication between devices over a VPN (Virtual Private Network).SSH (Secure Shell) is a protocol that is used to securely connect to and remotely control another device over a network. It is often used to securely connect to servers and other devices over the internet.AES (Advanced Encryption Standard) is a widely-used symmetric encryption algorithm that can be used to encrypt data. It is considered to be very secure and is used in a variety of applications, including securing data transmitted over the internet.Two-factor authentication adds an extra layer of security by requiring a second form of authentication, such as a code sent to designated device, in addition to password.A firewall can help to prevent unauthorized access to your devices by blocking incoming connections that do not meet your specified security rules.Avoid connecting IoT devices to unfamiliar networks, as they may not be secure.Usage of a secure internet connection, such as a wired Ethernet connection, to help prevent IoT devices from being accessed over an unsecured wireless connection.

Another group of concerns that are used in relation to IoT is the accuracy and reliability of IoT-based water quality monitoring systems. In order to prevent them, calibration, quality control, personnel training, redundancy, and data analysis should be taken into account when designing the system. To ensure accurate and repeatable measurements of water quality, it is important to regularly calibrate the sensors and other equipment used in the IoT system. This can be done using standard reference materials or by comparing the readings to those from a known reference system. By regularly calibrating the sensors, we can ensure that the data collected is reliable and accurate.

Implementing quality control measures, such as regularly checking the system for proper function and performing periodic maintenance, can help to reduce the likelihood of false alarms and ensure reliable data. IoT technology can be used to monitor water quality in real-time, which is critical for maintaining a safe and clean water supply. By continuously monitoring water quality parameters, such as pH, temperature, and turbidity, we can quickly identify when there are changes that could indicate contamination. By performing regular maintenance on the system, we can ensure that it is functioning properly and that the data it is providing is accurate.

Providing training to personnel who will be handling and maintaining the system can help to ensure that it is being used and maintained properly, which can improve accuracy and reduce the likelihood of false alarms.

The installation of multiple sensors or the use of backup sensors can help to improve the reliability of an IoT system by providing redundant measurements. This is because having multiple measurements from different sensors can help to identify potential problems with the system. For example, if one sensor detects a change in water quality, this change can be verified by other sensors before an alarm is raised. This helps to reduce false alarms and improve the overall reliability of the system.

Analyzing data collected by an IoT system can help to identify trends and patterns that may indicate a problem with the equipment or the accuracy of the measurements. This is especially useful in the case of water quality, as changes in trends could indicate contamination. By analyzing data over time, it becomes easier to spot these changes and take appropriate action.

## 6. Potential Efficiency of an IoT System 

Many ways can be used to improve the efficiency of water quality monitoring and management using IoT (Internet of Things) technology. 

Water quality parameters can be continuously monitored by IoT sensors, such as pH, temperature, and conductivity, providing real-time data that can be used to identify trends and potential problems.

IoT sensors can be incredibly helpful in detecting potential problems and ensuring a timely response. For example, if a sensor detects that a particular threshold has been exceeded, it can send an alert or notification to the appropriate personnel. This can help avoid potential disasters and ensure that any issues are dealt with promptly. IoT sensors can thus be extremely valuable in a variety of settings and scenarios.

IoT sensors are increasingly being used to collect data on a variety of factors that can impact equipment performance. One area where this data can be extremely helpful is in predicting when maintenance will be needed on equipment such as pumps or filtration systems. By understanding when these pieces of equipment are likely to fail, steps can be taken to reduce the likelihood of failures and improve efficiency. The data collected from IoT sensors can help identify trends and patterns that predict when maintenance will be necessary, ensuring that the equipment is always operating at peak performance.

IoT sensors can help to monitor and control water usage in order to reduce waste and improve efficiency. By collecting data on water quality and flow rate, IoT sensors can provide valuable insights that can help to optimize water usage. For example, if a sensor detects that the water quality is poor, it can trigger a warning that could prompt users to take action to improve the quality of the water. Similarly, if a sensor detects that the flow rate is too low, it could trigger a notification that could urges users to turn off the tap. In this way, IoT sensors have the potential to play a significant role in reducing water waste and improving efficiency [16,18,21].

Data collected by IoT sensors can be used to inform water management decisions, allowing water resources to be better managed. This is especially important in areas where water quality is a concern. By understanding how much water is being used and what the quality of that water is, decision-makers can make informed choices about how to best use and conserve water resources. IoT sensors can also help identify trends over time, which can give insight into long-term patterns of water use and quality. This information can then be used to develop policies and practices that will improve the management of water resources [26].

## 7. Conclusions

IoT in water quality monitoring has definitely arrived, and it is only going to become more prevalent in the years to come. Due to the benefits that it offers, IoT can be used to improve water quality monitoring in communities and businesses. IoT offers a number of advantages when it comes to water quality monitoring. It can help to provide real-time data about water conditions. This is valuable information that can be used to make decisions about how to best manage water resources. 

IoT in water quality monitoring can help improve water quality in a number of ways. For example, IoT can be used to monitor water quality in real-time, which can help identify potential problems early on. Additionally, IoT can be used to track long-term trends in water quality, which can help identify larger problems that need to be addressed. Finally, IoT can be used to automatically collect data from a variety of sources, which can help provide a more complete picture of water quality.

There are a number of challenges that need to be addressed before IoT can truly revolutionize water quality monitoring. For example, the accuracy of sensors needs to be improved, and data needs to be properly standardized. Additionally, the costs associated with implementing IoT solutions need to be reduced. However, there is no doubt that IoT has the potential to transform water quality monitoring for the better.

The potential benefit of using IoT systems for measuring and monitoring water quality can be building social awareness of what IoT really is and investigating what the water quality is and what it really consists of. The term water quality itself is associated with many components that, thanks to IoT systems, can be measured in real time and observed by a larger group of recipients—including people not technically related to both concepts.

## Figures and Tables

**Figure 1 sensors-23-00960-f001:**
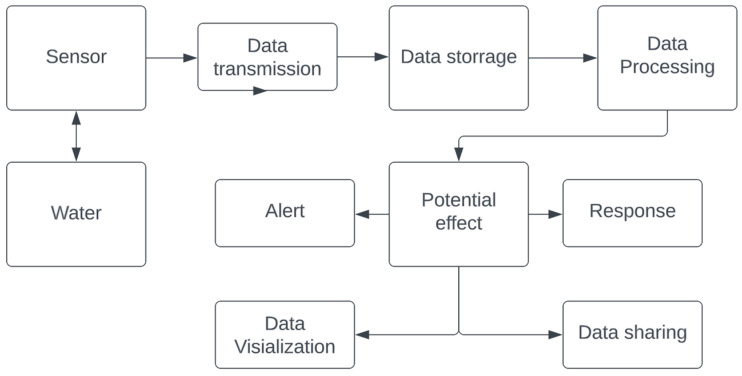
Example of an IoT system (a concept by the authors).

## Data Availability

Not Applicable.

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
