# Peer review of "IoT in Water Quality Monitoring—Are We Really Here?"

_sensors, 2023, doi:10.3390/s23020960_

Round 1

Reviewer 1 Report

The manuscript with the title “IoT in water quality monitoring - are we really here?”, which I reviewed, addresses this complex issue, sometimes too discursively. The merit is to focus the use of this network to monitor water quality, even if technically there are gaps, for example, there is no list of parameters to measure and in which contexts they should be measured (water basin, distribution network, etc.). In fact, in many countries the management of water is entrusted to bodies, which do not allow the application of this system or access to it. For this reason, the revision made reference to the issues of information management, which makes the answer to the concluding question of the abstract problematic. However, to make the functioning of the IoT more understandable, a schematic figure (e.g. flowchart) should be inserted. As already mentioned, it would be better to specify in which context the measurements are to be performed and with which instruments (i.e. sensors, smart meters, etc.): a table could also be inserted. Also, additions to the system may be made in case of abnormal results. Regarding data security, more information should be given on applications that could help maintain data integrity and availability. The references support the text quite well, even if they should be rewritten according to the instructions to the MDPI authors. If the manuscript were integrated with what was requested by the reviewer, its quality would improve, which is why I propose major revision. See also the attached file.

Author Response

Thank you for all work committed to the review. All responses are in the attached file. 

Reviewer 2 Report

The article has the character of scientific journalism and presents the concept/possibility of using new technology. There are very general statements in the text. There is no discussion of examples of system efficiency. Keywords do not reflect the text's content. The article should be supplemented.

Author Response

Thank you very much for your valuable comments. We have included a brief discussion in the revised manuscript. After taking into account all the reviews, the article gained a different version.

Reviewer 3 Report

IoT in water quality monitoring - are we really here?

The article looks interesting, but it only discusses the existing techs and some information on the practical problems faced due to varying field temperature, accuracy repeatability and unskilled labour handling of such iot equips can be elaborated by the author. 

How these IoT based equips can be trusted in these platforms and how to avoid false alarms and current market avaiable equips and their charectersitics discussion can be discussed which is more essential for the researchers to know about the equip in detail. 

Author Response

Thank you very much for your time and suggestions for the manuscript. We believe that the corrections posted address the problem presented, but we feel that they have enriched our manuscript. Thank you very much!

Added -  from line 384 to 413. 

Reviewer 4 Report

The manuscript entitled “IoT in water quality monitoring - are we really here?” presents review paper giving the information about the importance of using IoT in water quality monitoring. The topic is interesting and important for scientific audience. However, the structure of manuscript needs to be improved. The last sentence in introduction need to clearly define which is the aim of manuscript. The whole manuscript is written in the first and in the second person instead in the third person.

Please rewrite the first paragraph and the last sentence in the Conclusions to sound scientifically.

Page 2, line 63: Please, delete references from the title of section (21- 25, 35, 37)

Author Response

Thank you very much for your time and suggestions for the manuscript. We have made corrections based on your suggestions.
We hope that the manuscript has benefited from all the revisions.

Best Regards

Round 2

Reviewer 1 Report

The manuscript “IoT in water quality monitoring - are we really here?” that was returned to me has definitely improved in content. The authors have considered all comments and suggestions. Furthermore, some analytical difficulties encountered in the previous revision have been made more readable and understandable. It is believed to publish it in this version without further modifications.

Reviewer 3 Report

Can be accepted in this form

Reviewer 4 Report

The revised manuscript can be accepted as it is.